# The 3Rs in Experimental Liver Disease

**DOI:** 10.3390/ani13142357

**Published:** 2023-07-19

**Authors:** Sebastian Martinez-Lopez, Enrique Angel-Gomis, Elisabet Sanchez-Ardid, Alberto Pastor-Campos, Joanna Picó, Isabel Gomez-Hurtado

**Affiliations:** 1Instituto ISABIAL, Hospital General Universitario Dr. Balmis, 03010 Alicante, Spain; sebastian.martinez@goumh.umh.es (S.M.-L.); enrique.angel@alu.umh.es (E.A.-G.); joanna_pc@hotmail.com (J.P.); 2Departamento de Medicina Clínica, Universidad Miguel Hernández, 03550 Sant Joan, Spain; 3CIBERehd, Instituto de Salud Carlos III, 28220 Madrid, Spain; esanchezar@santpau.cat; 4Servicio de Patología Digestiva, Institut de Recerca IIB-Sant Pau, Hospital de Santa Creu i Sant Pau, 08025 Barcelona, Spain; 5Oficina de Investigación Responsable, Universidad Miguel Hernández, 03202 Elche, Spain; albertopastor@umh.es

**Keywords:** replacement, refinement, reduction, liver, research, disease

## Abstract

**Simple Summary:**

This article provides a review of recent studies that explore the application of the 3Rs in experimental animal models of liver disease, comparing different models for a correct search for replacement, refinement, and reduction methods. However, the limitations of each of the replacement techniques are identified, which highlights that although the use of animal models is still necessary, their number can be reduced and adjusted to the scientific question of interest, always taking into account their welfare and using alternative techniques to answer more specific questions.

**Abstract:**

Patients with cirrhosis present multiple physiological and immunological alterations that play a very important role in the development of clinically relevant secondary complications to the disease. Experimentation in animal models is essential to understand the pathogenesis of human diseases and, considering the high prevalence of liver disease worldwide, to understand the pathophysiology of disease progression and the molecular pathways involved, due to the complexity of the liver as an organ and its relationship with the rest of the organism. However, today there is a growing awareness about the sensitivity and suffering of animals, causing opposition to animal research among a minority in society and some scientists, but also about the attention to the welfare of laboratory animals since this has been built into regulations in most nations that conduct animal research. In 1959, Russell and Burch published the book “The Principles of Humane Experimental Technique”, proposing that in those experiments where animals were necessary, everything possible should be done to try to replace them with non-sentient alternatives, to reduce to a minimum their number, and to refine experiments that are essential so that they caused the least amount of pain and distress. In this review, a comprehensive summary of the most widely used techniques to replace, reduce, and refine in experimental liver research is offered, to assess the advantages and weaknesses of available experimental liver disease models for researchers who are planning to perform animal studies in the near future.

## 1. Introduction

The liver is a very important organ in organic homeostasis, with different functions, such as maintaining plasma glucose and ammonia levels, drug detoxification, bile synthesis, and storage and processing of key nutrients [1]. The hepatic response to insults like alcohol, infections, drugs and toxins, cancer, obesity and metabolic syndrome, genetic diseases, and autoimmune conditions is liver fibrosis, like wound healing [2]. Cirrhosis represents the end-stage of any chronic liver disease when liver acini are substituted by nodules [3] characterized by vasodilatation and generalized hypotension, related to portal hypertension, mainly due to increased intrahepatic resistance [4]. Patients with decompensated liver cirrhosis have a poor prognosis [5] and increased mortality, most often attributed to direct complications resulting from the loss of liver function, portal hypertension, and the development of hepatocellular carcinoma and, as such, has been estimated to account for one million deaths worldwide per year [6].

Experimental animal models have been used extensively to understand the underlying mechanisms of human disease, particularly liver cirrhosis. One of the principal reasons for their use is that animals and humans have many anatomical and biological similarities [7]. Throughout the 20th century, there have been great advances in biomedical sciences (invention of antibiotics, new methods of diagnosis and treatment of diseases, surgical techniques, the improvement of vaccination) thanks mainly to animal models, whose use in recent decades has increased exponentially, saving millions of lives and significantly increasing life expectancy. Currently, in Europe, the most used animal species according to the last report of EU Commission (Commission Staff Working Document) are rodents (mouse: 48.9% and rat: 8.4%), domestic fowl (5.3%), rabbits (4.3%), zebrafish (3.5%), guinea pig (1.4%), amphibians, cephalopods and reptiles (0.5%), dogs, cats, and non-human primate models (0.2%). Nevertheless, rodents are the most used for various reasons: Their high resistance to successive inbreeding, decreasing genetic variability between individual animals, their rapid reproduction rate, their small size and ease of handling, and their low cost in terms of accommodation and maintenance. Besides, mice are the species of choice used for geneti engineering and associated basic research. Notably, it is crucial to acknowledge as well that mice are not humans, and depending on the process under study, key differences between human and murine biology may affect our results [8,9], thus, being aware of them would allow us to better implement the 3Rs in our research (Figure 1).

Animal models of liver disease can be induced by different approaches [10]: (a) Oral or intraperitoneal (IP) administration of chemical compounds causing a direct injury and inflammatory reaction in the hepatocytes (CCl_4_ [11], thioacetamide [12], dimethylnitrosamine [13], dioxin [14], sodium arsenate [15], and ethanol [16]), (b) special diet causing non-alcoholic fatty liver disease (NAFLD) and cirrhosis, such as choline-deficient, L-amino acid-defined, methionine-deficient diet [17,18,19], and high-fat diet [20], (c) surgery, like bile duct ligation (BDL) [21], leading to cholestasis, infiltration of inflammatory cells in the portal area and liver fibrosis, or (d) fibrosis induced by deposition of immune complexes in the portal area and around the central vein area (concanavalin A [22] and xenogenic serum [23]). The resulting liver disease is usually presented as a painless clinical picture, with jaundice (especially in cholestatic models like BDL) and brightly colored urine. Jaundice is easily identifiable in albino strains by yellowing fur or ears, but in pigmented animals (black or agouti strains), it is not always immediate to detect. The color change in urine is also not easy to detect and requires the use of metabolic cages or using white bedding. In models of liver damage, early detection of the appearance of cirrhosis or of any color change in the urine could serve as endpoint criteria, but it is sometimes difficult to ascertain despite daily monitoring of the animals. Additionally, in advanced cirrhosis, ascites may appear in the animal (excessive accumulation of fluid in the abdomen). In most animal models, the onset of ascites is used as the endpoint to prevent animals from experiencing pain, stress, or discomfort [24].

In 1959, Russell and Burch [25] developed what was the first idea of good practice in animal research (that, over time, has been expanded with other Good Research Practice guidelines not specific to animals): The concept of the 3Rs (reduce, replace, and refine) to reduce the number of animals used, to look for an alternative for total or if not partial replacement, and refine the techniques used to minimize the pain or suffering of animals as much as possible. However, scientists have not stopped there, and new proposals have been developed based on Carol Newton’s three Ss (3Ss) [26], adapting them to what is called “the Three Cs” (3Cs) [27]: Full science, objective criteria, and culture of care. Nowadays, the pillar of any research that is carried out, even before starting a study, is to find the best way to do it, trying, whenever possible, not to use experimental animals. However, today, studies on experimental animals are still irreplaceable. It seems, however, that researchers are at the beginning of a new stage, and investigators are closer to replacement thanks to industrial development and research in the search for new biomaterials, computational development and its greater accessibility, and the use of artificial intelligence (AI), which will very probably change in a few years the vision of science as the community has seen it up to now. Meanwhile, governments, as well as the research community itself, have committed to work together for a responsible science where those studies that require animals will focus on an Integral Science: Proper scientific methodology, honesty, adherence to regulations; Objective Criteria: Retrospective evaluation harm-benefit analysis (3Rs: reduction, refinement, and replacement); and Culture of Care: bioethics, animal welfare, and responsibility.

**Figure 1 animals-13-02357-f001:**
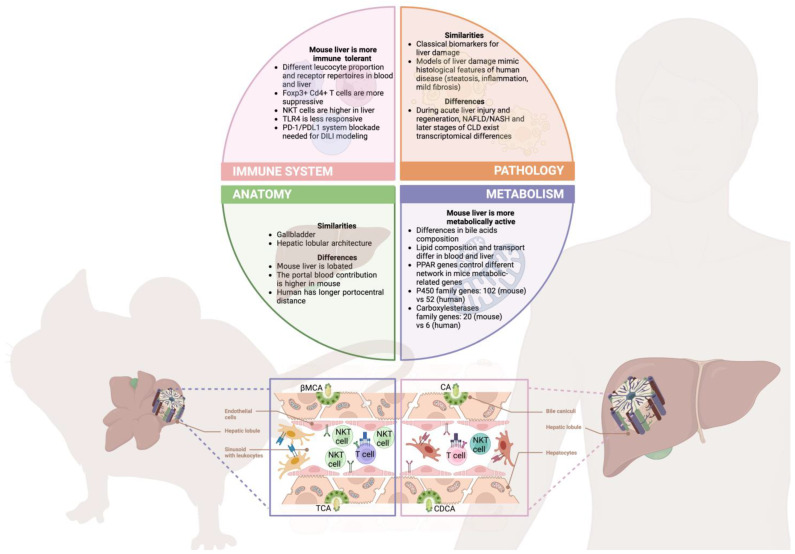
Relevant human and mouse differences in liver physiopathology. Although mice remain one of the best approximations to the human systems, key differences in liver biology need to be considered while designing projects in order to avoid failed experiments and inconclusive results, thus meeting the 3Rs principles. With respect to liver anatomy, albeit the external appearance differs between mice and humans, the functional microscopic lobule architecture is quite conserved in both [28]. This aspect translates into similar phenotypic manifestations during liver pathology (steatosis, inflammation, and fibrosis), although the symptomatology of advanced human disease is difficult to mimic in mice [29]. This might be attributed to the differences in rodents’ local immune system and metabolism. Regarding liver immunology, mice are considered more tolerant, with a reduced response to bacterial products and potent regulatory lymphocytes compared to humans [30,31]. Metabolism is considered to be more active in murine models, being exemplified in the expansion of metabolic enzyme genes and their respective increased production [32,33]. Remarkably, these distinctions are further exemplified in omics analyses that highlight partially divergent genomic responses to liver injury and disease progression [34,35], ultimately indicating that improved models, such as humanized mice, could reduce the gap between the two species [36]. This image was created using Biorender platform.

This review aims to help researchers, mainly those who work with experimental models of liver damage, to be aware that sometimes the animal model can be replaced by alternative techniques and the importance of the 3Rs in the day-to-day of their work with experimental animals.

## 2. The 3RS from an Institutional Point of View

Advances in scientific research have prompted the search for alternative methods in animal models of liver damage, with the aim of promoting their replacement, reduction, and refinement. Institutionally, university ethics committees and their research compliance offices play a key role in promoting and overseeing these approaches.

European legislation (Directive 2010/63/EU of the European parliament and of the council) is clear in the 3R’s approach: When there is a validated alternative method that replaces, reduces, or refines the use of animals, its use is mandatory. Existing validated replacement methods are the main reason for an unfavorable evaluation of an animal research project. However, not only in Europe, in the United States, the Animal Welfare Act (AWA) and its associated regulations [37] serve as the primary framework for overseeing animal research. According to these regulations, research protocols involving animals must undergo review and approval by Institutional Animal Care and Use Committees (IACUCs). The IACUCs play a crucial role in ensuring that the principal investigator has thoroughly considered alternative methods for proposed research activities involving animals. Similarly, in Japan, the Act on Welfare and Management of Animals [38] establishes the framework for animal research oversight. This regulation emphasizes the importance of considering the appropriate use of animals in research and encourages the exploration of alternative methods that minimize the reliance on animal use. It also highlights the goal of minimizing the number of animals involved in research whenever possible.

The replacement of traditional animal models implies the use of non-invasive techniques, such as cell cultures and in vitro models, which allow the mechanisms of liver damage to be studied more accurately and ethically. These in silico and ex vivo alternatives, based on the use of cell lines or isolated tissues, considerably reduce the need for animal experimentation and offer relevant results for understanding liver pathophysiology.

The reduction focuses on minimizing the number of animals used in experiments and optimizing research protocols to obtain the maximum information with the least possible impact. All kinds of institutional ethics review boards for animal research projects and offices for research compliance encourage the implementation of strategies that make it possible to achieve scientific objectives with a smaller number of animals through a rigorous experimental design and the application of appropriate statistical techniques. Current international regulations (European Union, USA, and Japan) do not require a biostatistician to be part of an ethics committee, but the existence of this figure should be mandatory as it is key to correctly applying the r for reduction.

Refinement involves improving experimental conditions to minimize pain, stress, and suffering in animals used in models of liver damage. Ethics committees and research compliance offices promote the adoption of measures that guarantee animal welfare, such as the use of adequate anesthesia, constant monitoring of vital signs, and the implementation of non-invasive sample collection techniques. Animal welfare is a crucial concern and is currently the best attended R worldwide since current legislation does regulate the existence of the attending veterinarian (USA research facilities and their IACUCs) [37], designated veterinarian (European Union breeders, suppliers and users of animals for research) (Directive 2010/63/EU), the person responsible for the welfare and care of the animals (Animal-Welfare Bodies in the European Union) (Directive 2010/63/EU), and the official in Charge of Animal Welfare (Japan) [38].

In summary, research institutions and research compliance offices establish various types of review boards to oversee animal research projects. These boards, with the help of workers with very specific professional profiles, play an essential role in promoting and supervising alternative methods in animal models of liver injury. Through the promotion of replacement, reduction, and refinement, it seeks to guarantee the scientific and ethical integrity of research while protecting animal welfare. These approaches contribute to moving towards a future where the reliance on animal experimentation in this crucial scientific field is significantly reduced and, possibly in the longer term, completely replaced.

## 3. Animal Models in Liver Disease

The animal species used for animal models of cirrhosis depends on the main objectives of the study [33], but the most commonly used are rodents, especially mice and rats. Mice are mainly used if the primary goal of the study is organ harvesting to assess liver fibrosis and inflammation, and where complex surgical interventions are not required. In mice, due to their small size, surgical interventions are more complicated. Although a high degree of technical expertise is always necessary, in mice, an ocular microscope is sometimes needed and is not always available. Thus, in surgery models, rats are used more often since reproducibility is much higher, and mortality is lower [33].

Within each species, there are also differences between strains. In mice, for example, there are interstrain differences in the development of diet-induced NAFLD between C57BL/6, BALB/c, and C3H/HeN mice [39,40]. There are gender-specific differences too, for example, in the development of steatosis in rats [41]. Male animals are generally used to eliminate potential confounding factors, such as the complex female hormonal status, but it has detrimental consequences for women’s health [42], so the legislation ultimately requires detailed experiments performed on both sexes. Therefore, each model has specific advantages and disadvantages and must, therefore, be chosen according to the research questions to be answered.

## 4. 3RS in Experimental Models of Liver Disease

### 4.1. Replacement

Since the publication of Rusell and Burch’s guide for reduction of animal pain in 1959 [25], the application of less aggressive handling and experimentation techniques to animal research in the form of the 3Rs principles has permeated every aspect of scientific research, including its legislation [43]. These principles have their major (European) legislative acknowledgment in Directive 2010/63/EU, which states that “An experiment shall not be performed if another satisfactory scientific method of obtaining the result sought, not entailing the use of an animal, is reasonably and practicably available”.

Owing to the variability regarding etiology and pathophysiology of liver diseases, a plethora of animal models have been developed over the years in order to adequately study these pathologies [29]. In addition, a wide variety of in vitro and in silico approaches have concurrently been engineered with the aim of replacing animal use in hepatic research (Table 1). Those replacement alternatives will be explored in this section.

#### 4.1.1. 2D In Vitro Liver Models: Monocultures and Cocultures

In general, the most widely used method for studying cell biology is the traditional 2D monolayer culture, where isolated cells are seeded in an appropriate culture medium over a flat, stiff polystyrene surface. In the field of liver fibrosis, Hepatic Stellate Cells (HSC) cultures have been usually used for high-throughput screening of fibrosis-related compounds and potential treatments [44,45]. Classic in vitro HSC cultures are based on stellate cells transdifferentiation, transitioning from a quiescent, high-lipid, healthy liver phenotype cells to a myofibroblast-like, activated phenotype, typical of a diseased liver [3,44]. 2D hepatocyte monocultures (commonly hepatocellular carcinoma-derived cell lines) have also been extensively used in different liver pathology studies, such as NAFLD. In these models, steatosis is induced in cultured cells by the addition of a mixture of free fatty acids to the culture medium [46].

However, the aforementioned strategies both fail to recapitulate the complex intercellular interactions characteristic of hepatopathies, the genomic and proteomic changes that cells undergo, and their repercussion in cytokine and extracellular matrix (ECM) production. Despite their utility, classic monocultures are of limited use for studying liver diseases, as these strategies do not consider intercellular interactions between cell types. Bearing that in mind, coculture systems have been used in order to explore communication between liver cells, investigating the intense crosstalk established by means of cytokines, growth factors, chemokines, reactive oxygen species, and plasma proteins, therefore, allowing researchers to measure changes in gene/protein expression and functional/physiological effects [3,44].

Even though this approach is unable to differentiate between the effects of one cell type over the other, it identifies the overall effect of cell interaction. If the experimental design involves coculture of non-adherent and adherent cells, both populations can be cultured together, washing the non-adherent one before sample analysis, thus measuring specific adherent cell changes. Following this strategy, transwell inserts offer the possibility of establishing indirect cocultures, physically separating both cell types but allowing secreted factors exchange. A halfway point between coculture and paracrine cellular interaction pathways involves the stimulation of a cell type with the conditioned medium of another cell culture, containing all soluble factors, exosomes, microparticles, and cytokines produced [47,48,49].

Collagen overexpression and its accumulation, forming liver scar tissue and fibrous septa, are the most characteristic events of the disease observable in ECM, being a hallmark of the progression of chronic liver disease. ECM proteins and their interaction with the cells condition cellular behaviors, such as cell morphology and gene/protein expression, by means of the activation of multiple signaling pathways [50,51]. Interestingly, research has shown differences in the stiffness of the surface the cells are grown into activate those crucial signaling pathways. For instance, primary HSC cultures seeded on supports of different stiffness adopt different phenotypes depending on it (12 kPa, extremely cirrhotic rat liver; 0.4 kPa, healthy rat liver). These values obviously contrast with the approximately 10,000 kPa stiffness of a regular culture dish, bringing to light the inaccuracy of the most common in vitro supports in which experiments are carried out [45,52]. In vivo, HSCs are subjected to different mechanical/pressure forces depending on their location (portal ducts are stiffer than pericentral regions, etc.) [44,53], thus showing a heterogeneous profile that is very difficult to imitate in vitro. In addition, elegant research has demonstrated that mechanical stress increases TGF-β mRNA and protein expression levels, inducing an epithelial-mesenchymal transition in HSCs [54,55].

Taken together, these studies show that reductionist 2D monoculture does not fully address the complex multicellular processes and mechanical heterogeneity that shape the healthy fibrotic liver, being unable to accurately recapitulate the physiological interactions in which cytokines, nutrients, oxygen, and pressure shape cell behavior.

#### 4.1.2. 3D In Vitro Liver Models

It is clear that highly accurate in vitro liver studies require a precise recreation of the liver microenvironment, taking into account inter-cell interactions, paracrine signaling, and secreted mediators. Hence, 3D in vitro models represent the next logical step towards mimicking in vivo conditions as closely as possible, allowing researchers to construct complex microscale ultra-structures for cultures. When achieved, these 3D strategies can provide a fast, high-throughput, accurate liver culture model for hepatic research [56] (Table 1).

Cell Stacking: The most conceptually simple strategy for three-dimensional cell engineering uses monolayer cultured cells in the “height” dimension. Cell sheet stacking undertakes this approach by using temperature-responsive culture dishes coated with polymers (poly(*N*-isopropylacrylamide) (PNIPAAm)) that change their hydrophilicity, hence the adherence of the cells, depending on the temperature. This allows the cells to be harvested when the hydrophilic conditions are met (20 °C) without any damage to membrane proteins. The preservation of intercellular interaction allows the stacking of several cell sheets and the reproduction of a more complex 3D structure [57,58]. In hepatic research, the combination of both parenchymal and non-parenchymal liver cell sheets has emulated some liver functions, such as albumin and urea synthesis. However, the absence of oxygen supply to the inner zone of the 3D-engineered tissue due to the lack of vascularization conditions limits cell viability to a short period of time that ends in an ischemic event [59,60].Spheroids as a 3D in vitro hepatic system: Spheroids are cellular clusters that can include one or more cell types and that typically acquire a spheroidal structure in the presence or absence of a matrix, recapitulating the hepatic in vivo 3D environment, being the 3D spheroid system applied to hepatocyte culture one of the first models of this technique developed [61]. Hepatocyte spheroids promote intercellular and ECM–cell interactions, the development of metabolites, gas, and nutrient gradients across the spheric-like structure and have been demonstrated to imitate and maintain liver-specific functions [62]. This approach has probed its utility in the study of drug pharmacokinetics and toxicity tests, making feasible its application for the determination of an appropriate drug dose [63].Current hepatic 3D spheroid-assembling methods have evolved [64,65], giving rise to a wide variety of protocols, being some of the most used the micromolds of non-adherent materials [66], rotation of cultured cells on 96-well non-adherent plates [61], stirred bioreactors [67], and methods that depend on gravitational aggregation of cells using hanging drop cultures. The availability of commercial solutions for hanging drop culture systems (ED Biomatrix. Ann Arbor, MI, USA; InSphero. Schlieren, Switzerland) has eased an initial standardization strategy for the technique [68].A variable that clearly limits the standardization of spheroids includes their non-homogeneous size, which directly influences experiments’ reproducibility [63]. Several technologies have investigated the generation of standard homogeneous spheroids in order to overcome this limitation by means of micromold-based spheroids [69], or by inoculating primary hepatocytes into pre-cultured 3D spheroidal nylon-scaffolds of non-parenchymal cells producing size-consistent particles with hepato-specific functions [70]. Even though there are some commercial solutions for spheroid assembly, their cost and difficulty make them unattractive for their implantation in most laboratories. On top of that, the clustering systems previously defined share several common limitations with monolayer cultures, such as the absence of a standardized culture scaffold stiffness and topography [71].3D matrix/scaffold-based cultures: The use of 3D scaffolds for 3D culture of hepatic cell populations is a frequently used system just by itself or in combination with other technologies (such as its application in homogeneous spheroid generation explained in the previous section). Culture matrixes can be developed out of synthetic (scaffolds synthetic polymers [72]) or biological materials biological sources (naturally derived hydrogels: Alginates, celluloses, or polyethylene [73]). Synthetic materials are extremely flexible and can be easily fabricated, but their microarchitecture and biomechanical behavior, key properties for proper cell culture, are not adequate. The current strategy to overcome these limits is based on the development of 3D scaffolds from biological ECM-derived materials, such as basement membrane or type-I collagen. These ECM-derived gels are characterized by the preservation of cell–ECM interactions. Despite that, these biomaterials also have several limitations, mainly the heterogeneity of mechanical properties between different batches of gels and the incapacity of resembling physiological liver 3D microarchitecture [74]. In those conditions, cultured cells encapsulate randomly in ECM-derived gel scaffolds, randomly self-organizing into not natural liver microstructures and also not reproducing and/or maintaining specific liver functions over time due to the absence of tissue-specificity of these biomaterials [44].3D bioprinting of liver tissue: 3D bioprinting includes a group of techniques for the fabrication of three-dimensional, cell-packed, liver-like structures with a determined geometry by the manipulation of a combination of biological materials, biochemicals, and living cells, usually called “bioinks” [74]. Based on their handling, the most common printing technologies can be classified into three different categories: Droplet-based, photocuring, and extrusion-based bioprinting [45,74].
Droplet-based bioprinting also known as inkjet-based bioprinting, was the first 3D bioprinting technique developed. Originally proposed by adapting a common inkjet printer to work with bioink-loaded cartridges, current droplet-based bioprinters use an actuator to deposit droplets of biomaterial onto a substrate to generate a 3D structure [75]. In addition to their simplicity and commercial availability, droplet-based bioprinters still remain the first choice for applications that require the printing of a high accuracy pattern, being able to even generate single-cell droplets of bioink, thus building constructs with a similar resolution to cellular dimension [76]. However, their popularity is being progressively reduced due to the overstress that is generated in the cells by the thermal and piezoelectric actuators that form the printer [77].Photocuring bioprinting: Currently known as laser-based bioprinting due to the implementation of this lighting technology to treat photo cross-linkable bioinks, this approach can generate highly accurate 3D structures with a resolution of up to 50 µm [45]. Laser-based bioprinting is especially recommended for high-fidelity applications at low resolutions (i.e., vascular network ultrastructure printing [78]). Photocuring bioprinters can effectively mimic the lobular vascular network from a technical, structural point of view. However, the use of this technology strongly limits the variety of bioinks available due to the necessity of photo-crosslinking properties or the use of functionalizing reagents [78]. In combination with their low throughput, the cell-unfriendliness of the method, and the absence of commercially available systems, laser-based bioprinting is not a standardized 3D modeling method yet [45].Extrusion-based bioprinting: It is based on the extrusion and deposition of a bioink filament on the printing surface [77]. Depending on the mechanical approach by which the biomaterial is extruded, three different subcategories are identified: Pneumatic extrusion (the most widespread, controls the flow of ink and pressure through an air compressor), extrusion with a syringe pump (controls precisely the extruded volume of bioink), and screw extrusion (the least popular due to the enormous stress that screw rotation places on the cells) [74]. Extrusion-based bioprinting is currently the most popular technology for liver tissues. The main reason is that it overcomes the cellular stress and damage generated by droplet-based approaches [77].As stated before, there is a wide variety of bioprinting technologies, with their respective market approach, features, price, and technology. Conversely, the design, standardization and optimization of tissue-specific bioinks is still a challenging task in this field [79]. The materials used for bioprinting need to be cell-and printing-process compatible and provide the mechanical and functional properties that mimic liver tissue, thus allowing maintenance of hepato-specific cell behavior in long-term cultures [46]. Private companies have developed some commercially available solutions for bioprinting materials, such as Insphero^®®^, whose extrusion-based 3D spheroids of hepatocytes, HSCs, and Kupffer cells were printed in liver ECM-hydrogels. Another example is the Novogel 2.0 from Organovo^®®^, which has been used to extrusion-bioprint HSCs/HUVEC cocultures allowing long-term maintenance of cells [45]. Nevertheless, the use of proprietary printing techniques dramatically increases costs, and the need for a complicated experimental setup restricts its accessibility to the research community.
Decellularized Liver scaffolds: Decellularization of liver scaffolds is an elegant strategy to overcome the macro- and microstructure-related issues derived from previously mentioned 3D culture techniques, where liver microanatomy could not be fully reproduced. Decellularized liver scaffold aims to provide a 3D structure that has the optimal conditions for hepatic cell culture survival and proliferation while preserving the intrinsic organ vascular network [80]. This approach is based on eliminating parenchymal and non-parenchymal populations of the liver by perfusing the organ with decellularizing solutions (i.e., detergents, delipidating enzymes, and nucleic acid-degrading enzymes) while preserving three-dimensional ECM structure [45]. Decellularized livers can then be used as 3D ECM bio-scaffolds for recellularization and tissue engineering [44]. A key advantage of decellularization-recellularization protocols lies in the preservation of liver-specific ECM macro- and microstructure and crucial bioactive molecules for ECM-cell interactions, thus assuring the necessary stimuli for hepatic cells engraftment, survival, and physiological behavior, consequently allowing to overcome other 3D platforms’ limitations [80].Some drawbacks of these culture models include the fact that decellularizating perfusion protocols must be tissue-, disease-, and species-specific, mainly due to the significant variations of cell populations abundance and proportions and ECM protein composition that characterize healthy and diseased liver and could affect the recapitulation of 3D environment during cell elimination [81]. Another limitation of the strategy is the limited availability of human liver tissue for the optimization of the model (and the technical complication of human liver perfusion, forcing researchers to adopt alternative tissue in vitro incubation methods). Hence, most liver decellularization studies have been performed in rats, even though successful decellularizations have been reported of whole sheep and pig livers [80].Precision cut liver slices (PCLSs): PCLSs are a versatile, ex vivo model that maintains complete liver architecture and populations (including infiltrating immune cells), therefore, avoiding the limitations of previous models when it comes to macro- and microarchitecture mimicking, as it is a genuine slice of tissue [82]. Among their main benefits, their reproducibility, low cost, and high overall cell viability should be highlighted, allowing researchers to perform experiments in cultured PCLSs for 5–7 days (although this relatively short time could be considered a drawback of the system) [82]. In addition, the combination of PCLSs with microfluidic devices has boosted the in vitro evaluation of multi-organ interaction and culture technology [45].Nevertheless, access to fresh human liver tissue (the most valuable for research) is one of the pivotal limitations of PCLS: The scarcity of healthy human liver biopsies in combination with the coordination between surgeons, clinician teams, and researchers strongly limits its use [82]. Another drawback is the lack of biopsies on healthy tissue, forcing researchers to work with distal portions of the diseased liver that are not actually healthy. The same situation happens with diseased tissue, as research usually needs a clearly defined background to proceed and identify interesting outcomes. However, harmed liver biopsies are extracted from patients undergoing different treatments/therapies and might have a combined diseased background with metabolic, alcohol/drug intake, cancer, or genetic components [83]. Bearing that in mind, the main alternative for liver slice obtention are animal organs, which, unfortunately, have shown different proteomic profiles and outcomes, depending on the species [44]. Lastly, a clear disadvantage of PCLSs independently of human or animal sources is that slice preparation inevitably generates two cut-injured surfaces that undergo regeneration processes that result in HSCs activation [45].Liver-on-chip: As stated before, liver cells in vivo are put under constant and diverse stimuli. Organs-on-Chip, specifically Liver-on-Chip, are microfluidic devices that in vitro recapitulate the physiological properties of tissue microenvironment. Liver-on-Chip specifically considers fluid flows and 3D architecture of the hepatic sinusoid, allowing for 3D cell culture with a greater degree of in vivo-like environmental cues, highlighting fluidic shear stress as one of the key features of healthy and diseased liver 3D culture [84,85]. Liver-on-Chip devices can be classified in the following categories: Gravity-driven perfusion platforms, Pump-driven perfusion platforms, and 3D mass culture systems [85].
Gravity-driven perfusion platforms: They are the simplest version of these microfluidic devices, but also the ones that less resemble physiological conditions, relying on gravity to perfuse cultured cells. This approach eliminates the necessity of tubing, pump components, and power supply dependence, hence, flow rate, nutrient supply, and shear stress are not constant over time [86]. In this device, the tilting of the chip forces culture media to flow from one reservoir to another through microfluidic channels where liver cells are cocultured. The fluidic conductions are engineered so that the cells experience a unidirectional flow of media (by returning media with different connections to the original reservoir). Despite some new designs that try to avoid flow-related issues (bi-directional, recirculating organ-on-chip), the ease of use and resilience against bubbles make simple, gravity-pumped organs-on-chip a valuable and affordable option [86,87].Pump-driven perfusion platforms: They constitute a higher level of physiological similarity than the previous ones, in which peristaltic/syringe pumps deliver a high enough, consistent flow rate, thus assuring a better grade of nutrient supply and shear stress [88]. These systems allow researchers to design far more complex chips to study specific biological aspects of cells in culture, even combining 2D perfusion systems with a gradient generator to imitate the metabolic zonation of the liver [89] or the development of membrane-based, pump-driven devices for the coculture of primary human hepatocytes with hepatic non-parenchymal cells, successfully resembling albumin and urea secretion in comparison with classic culture techniques [90].3D mass culture systems: Designed to promote cell proliferation and aggregation into three-dimensional, self- or engineered-organized structures (following similar principles as those explained in the spheroids section of this paper) while putting cells under the flow conditions previously described. One of the best examples in liver disease is the device developed by Lee et al. [91]: A mimetic sinusoid-like structure formed by a wide inner channel for cell culture and an outer channel for culture medium flow. The microfluidic arrangement is connected by perpendicular channels that allow a continuously perfused 3D culture with limited shear stress that highly imitates the physical distribution of sinusoidal endothelium and even LSECs fenestrae.Unfortunately, the cost and complexity of these microfluidic systems, in combination with the absence of popular, commercial options, clearly limit the applicability of the technology for investigation. The vast majority of liver-on-chip and microfluidic organ-on-chip devices are developed by the research groups themselves, compromising their reproducibility [46]. In addition, the complex and diverse etiology of most liver diseases, in which not only the liver, but many other players, such as adipose tissue and gut, take part, bring the necessity of studying the interactions between multiple organs in these already complex devices. Even though multi-organ systems and body-on-chip approaches are starting to be a reality, these highly complicated microfluidic culture strategies are still more a field under research rather than a real, affordable, and accessible alternative to non-specialized research teams [92].Organoids: Organoids, as defined by Lancaster and Knoblich in 2014 [93] and Huch and Koo in 2015 [94], are an “in vitro 3D cellular cluster derived from tissue-resident stem/progenitor cells, embryonic stem cells, or induced pluripotential stem cells capable of self-renewal and self-organization that recapitulate the functionality of the tissue of origin.” From this general definition, liver organoids arise as three-dimensional culture systems developed by embedding hepatic stem, progenitor, or tissue resident cells in hydrogel matrixes that resemble ECM properties [46]. The beginning of organoid culture starts with the isolation of the aforementioned stem/progenitor/pluripotent cells from embryonic or adult tissues, which then are cultured and stimulated in media with growth factors and support matrixes that imitate endogenous, physiological signals that give rise to liver tissue during embryonic patterning [95]. Organoids completely outplay 2D and even some 3D culture models (Table 1) when it comes to (patho)physiological modeling, cell differentiation, interaction, and migration. In addition, their high genomic stability and easy culture expansion make them a highly suitable model for long-term storage and high-throughput compound screening [96].Even though many researchers have defined specific protocols and schemes for organoid generation, this technology is still extremely proprietary. The long, time-consuming protocols and particular growth factors, stimulant agents, and molecule cocktails required for each step of organoid generation keep this 3D culture method as a not widespread option [97]. However, no matter the strategy followed, the process of liver organoid formation (which obviously mimics physiological liver organogenesis) considers three key factors: Scaffolding/matrix material, signaling for cell differentiation, and starting cell types. Firstly, ECM stiffness, origin (biological or artificial), and composition have a great impact on the cell behavior and its metabolism, as previously commented. In organoid generation protocol, commercial Matrigel (a biological ECM extracted from Engelberth–Holm–Swarm (EHS) mouse sarcoma) is the most popular scaffolding material for liver organoids [98]. Although biological-source hydrogels better imitate the structure and microstructure of liver ECM while also being naturally embedded with a milieu of proteins, growth factors, cytokines, and signaling molecules that biologically prime organoids’ cells, their lack of a defined composition and consistency limit the reproducibility of assays and large-scale, commercial production [99]. Furthermore, the murine origin of matrigel restricts its applicability to clinical or human-based research processes.Secondly, cell signaling differentiation stimuli must be carefully implemented in culture media to prime liver development pathways and self-organization. Finally, the starting cell types play a pivotal role in the whole liver organoid formation process by directly influencing their evolution. Monoculture or coculture of different starting cells (primary/adult stem cells, embryonic stem cells, induced pluripotent stem cells, etc.) follow different differentiation paths, therefore requiring specific treatment [100]. Benefits notwithstanding, liver organoids remain an uncommon culture strategy for basic research groups. The time-consuming, low efficiency, and poor control of morphology and composition, in combination with the tendency to express immature fetal markers or differentiated cells, are obstacles that basic researchers have to overcome. The scarcity of primary liver tissue that conditions organoid development to immortalized or tumoral cell lines, a lack of exposure of liver organoids to gut microbiome, and the absence of optimized differentiation protocol on the industrial scale are the main issues that keep this technology as a promising but not yet applicable technology at mass, commercial production, or clinical applications [100,101].

#### 4.1.3. Computational Models

Computational models and in silico approaches have acquired great importance along with the evolution of computers and the rising relevance of data science, which allows researchers to investigate large databases in a high-throughput, fast manner. In silico strategies have been developed to study food–drug interactions, liver metabolism modeling, and fibrotic process simulation that recapitulate collagen deposition and inflammatory processes in central venous regions [102,103,104]. However, Drug-Induced Liver Injury (DILI) research stands out as one of the most benefitted areas when it comes to computational models. Chemical transforming and clearing via first-pass metabolism and detoxification reactions is the main task of the liver. Due to these metabolic functions, the withdrawal of many pharmaceutical products is related to the modification of chemical species carried out by the aforementioned hepatic metabolism. Therefore, it is easy to understand the broad interest and potential in developing in silico tools for preclinical and even basic computational testing of hepatotoxic effects, which allow researchers to avoid in vitro and in vivo unsuccessful experiments [105].

Different approaches have been designed to mathematically reproduce liver physiology and its modifications by xenobiotic interactions, disease progression, and other pathologies. Knowledge-based prediction algorithms are computer programs that can be trained with input and output data to determine the probability of a biological outcome or event. Then, by integrating real input cases, properly trained knowledge-based prediction algorithms can predict potential endpoints of hepatotoxicity or liver damage by preliminary properties of certain molecules, drugs, or substances in development [106]. A similar strategy is used by cheminformatics-based models. In this case, in silico models benefit from the structure–function relationships established for chemical and biological molecules, in a procedure named Quantitative Structure-Activity Relationship (QSAR). QSAR models relate molecular descriptors (molecular weight, number of carbon atoms, fingerprints, molecular patterns, etc.) with their biological activity and then determine possible outcomes [107,108]. Bioactivity-based models can improve structure-based technology by integrating biological data, such as gene expression profiles, biomarkers (mitochondrial damage, oxidative stress, etc.), or toxicologic information, to better predict the required endpoints [109,110]. Alternatively, expert knowledge approaches can use QSAR methods and code explicit decision rules that identify molecular patterns as related with a determined outcome/pathology [111].

Despite the fast improvement experienced by computational algorithms and models in recent times, their major drawback is the heavy dependence on the quality of training data. Even with the invaluable help of expert hepatologists and the implementation of explicit decision rules in models based on their knowledge, bigger, high-quality datasets (obtained from in vitro or in vivo experiments) are required for algorithm training [112]. In addition, the intricate and complex metabolic and biological pathways involved in liver physiology and the relatively limited knowledge of many of them keep computational models as a still-under-development technology [113].

**Table 1 animals-13-02357-t001:** Comparison of advantages and disadvantages of popular replacement strategies in hepatic research.

	Advantages	Disadvantages	Key References
2D Mono-/cocultures	Easy, reproducible handling High throughput	Intercellular interactions Macro-, microenvironment Mechanical heterogeneity of liver Reductionist view of disease	[44]
Cell stacking	3D environment Preservation of intercellular proteins	Short incubation time Absence of oxygen supply Ischemic events in long-term cultures	[59,60]
Spheroids	Intercellular and ECM interactions Metabolites, gas and nutrient gradients across spheroid	Non-standardized size Difficult reproducibility of experiments Elevated price	[61,62,63]
3D scaffold-based cultures	Allows better spheroid standardization ECM-cell interactions	Heterogeneous mechanical properties between scaffold batches Possibility of random, non-liver organizations of cells	[72,74]
3D Bioprinting	High versatility thanks to multiple bioprinting techniques	High cost of proprietary reactives Lack of ECM interactions	[45,75,78]
Decellularized liver scaffolds	Maintenance of liver macro- and microstructure ECM interactions Preservation of organ vascular network	Highly specific decellularization protocols with difficult reproducibility Limited source of human liver tissue	[75,80,81]
Precision cut liver slices	Complete maintenance of liver architecture and cell populations High reproducibility Low cost Combination with other technologies (e.g., liver-on-chip)	Scarring process at cut-injured surfaces Short culture time Limited source of human liver tissue Lack of healthy liver samples Precise coordination between clinicians and researchers required	[82]
Liver-on-chip	Good recapitulation of physiological conditions Shear stress, nutrients supply and waste clearance Metabolic zonation emulation	High cost of technology Absence of commercial/popular solutions Limited reproducibility Restricted diffusion due to complex experimental design and infrastructure	[85,86,87,88,89,90,91]
Organoids	Excellent (patho)physiological modelling Good cell differentiation, interaction and migration Culture expansion and high genomic stability	Non-standardized generation protocols Long, time consuming engineering process Highly complex experimental design and infrastructure Matrix-related limitations (murine Matrigel, batch effect, etc.)	[46,96,97,100,101]
Computational models	Exceptionally high throughput Technology under constant improvement Positive benefit/cost ratio Excellent coupling with further validations	Limited to existent physiological/expert knowledge Requires in vitro/in vivo experiments for validation Heavy dependence on training data quality	[102,104,106,111]

### 4.2. Refinement

Refinement in animal research refers to the process of changing experimental procedures or husbandry practices in order to minimize any potential for pain, suffering, distress, or harm to the animals involved. Since its original definition by Russell and Burch [25], the concept of refinement has evolved, not only including the time in which animals are being used but also focusing on alleviating any other adverse effects that animals may experience during their lifetime [114]. In that sense, the collective effort from researchers and authorities nowadays is directed towards reducing discomfort in animals as well as improving their quality of life, while maintaining the scientific rigor in experimental procedures.

In liver disease research, many aspects of the steps involved are susceptible to refinement. From less stressful handling animal techniques to modern methods of non-invasive imaging, several improvements have emerged to fulfill this purpose:

#### 4.2.1. Animal Husbandry and Monitoring

In recent years, there has been a growing recognition of the specific environmental needs of the different species used in research, as it can affect the wellbeing of the animals and the reproducibility of the results [115,116]. In the specific case of mice, as they are considered social animals, they should be housed in stable groups (European Commission Recommendation 2007/526/EC), which, in principle, would allow the animals to express their natural behavior. It is also true that group housing benefits humans for practical and economic reasons, given our needs for standardization and ease of sanitization [116]. In that sense, male housing is one of the aspects of husbandry still subjected to fine-tuning nowadays [117,118]. The social structures which free-living mice exhibit most typically in nature (one male, several females, and their offspring) cannot be replicated in the laboratory for practical reasons [119,120]. As such, males housed in groups display territorial behaviors, giving rise to social stress, violence, pain, or even death [118]. When this aggression happens, it may induce variability in the results and the appearance of artifacts detrimental to chronic liver disease research, such as wound-associated infections [121]. Spontaneous bacterial peritonitis, for example, is a characteristic and severe infection in the progression of advanced liver disease which constitutes a subject of study on its own [11,122]. Thus, keeping mice free from other infectious diseases would benefit the scientific validity of the results in this area.

It is also worth noting that lack of social interaction in the alternative of single housing for males could be as detrimental as the distress generated by the common aggressive behaviors displayed when caged in groups [117,123,124,125]. In consequence, it is crucial to assess which housing method is more suited to the needs of our experiments while having the welfare of the animals as a priority. Depending on the strain or the model investigators are working with, for instance, they could consider individual housing if avoiding the injurious fighting in aggressive strains [126,127,128] outweighs the stress associated with the lack of social interactions. Conversely, if it is needed to avoid the effects of isolation, reducing the animals per cage [129,130], as well as forming groups from littermates when possible [131], have been suggested as methods to ameliorate aggression. Other feasible strategies to implement are related to housing management. It is known that cage cleaning, although essential for the health of the animals, disrupts odor cues involved in the establishment of the social hierarchy of the group, promoting, as a consequence, aggression [132,133]. Accordingly, transferring nesting material from the old cage to the new one has shown to reduce aggression and stress [133]. Moreover, it is widely recognized that other adverse stimuli during handling, such as mouse identification by ear-based methods like ear-notching, can be painful and aversive for mice [134]. Although reliable alternatives, like tail tattooing, have emerged, ear-notching remains one of the most used methods due to its simplicity, permanence, and the possibility of using the removed tissue for genotyping [135]. Additionally, there is still controversy regarding the suitability and welfare cost of both, being tail tattooing also stressful and painful to mice [135,136]. Alternatively, if identification is meant for short-term purposes, using non-toxic fur dyes is considered a refined alternative to the methods discussed [137]. On the other hand, tail-based restraining techniques, albeit a common practice, are considered aversive as well, making the alternatives of cupping and tunnel handling [138] a great refinement method to reduce aggression [139]. As a matter of fact, cupping and tunnel handling have been shown to ameliorate the behavioral signs of stress compared to tail-based methods, even when full restraining is unavoidable for the procedure [138,140]. In the model of CCL_4_-induced cirrhosis [141], for example, an adequate immobilization of the animal is crucial during oral gavage of the compound to avoid the complications associated with this dosing route [142]. Thus, efficient non-aversive handling [143], together with the refined technique of three-finger scruffing during restraining, would greatly improve the welfare of the animals throughout gavage-based protocols [140,144].

Another successful method that has been shown to improve animals’ quality of life is environmental enrichment [117,145,146]. Although its definition is still not clearly defined, it can be referred to as any practice that provides the animal with means to express, at least partially, natural behaviors, such as exploring and foraging, thereby promoting their wellness [147]. The basic approach, now contemplated explicitly by the current codes of practice (2007/526/EC), usually consists in the addition of nesting material to the cage, as well as material meant for hiding and climbing [148]. Conversely, researchers should be cautious with additional increases in the complexity of the enrichment as it has shown mixed results [147]. Running wheels, for instance, have been proven to increase competitive behavior leading to aggression and stress [149,150]. As valuable items are susceptible to monopolization, especially under aggressive strains, complex structural enrichment should be tailored to each project individually.

To assess whether the measures mentioned above have an impact on the well-being of laboratory animals, several tools have emerged in order to monitor their health. Simple ‘Cage side’ behavioral indicators have been suggested as a helpful guide to examine their general condition routinely [118]. How mice build their nest and sleep or whether they are feeding enough can give us important hints about their comfort levels. When it comes to pain, more than a decade ago, the so-called “grimace scales” were proposed as a standardized coding system for mouse facial expressions triggered by noxious stimuli [151]. To this date, the system has been adapted for numerous species [152] and has been widely recognized as a potent tool for fast evaluation of suffering. Although in mice, it has mostly been used for scoring pain retrospectively, it offers the potential to support animal welfare by allowing researchers to intervene more precisely when needed [152]. Regarding anxiety, more sensitive tests, such as the elevated maze test and open field test, can be used to identify this adverse effect in mice when greater levels of stress are expected [153,154]. If further thorough approaches are needed, Home Cage-Monitoring stands out as a non-bias system for long-term monitoring of animals subjected to painful procedures and their response to analgesia [155]. While its implementation may pose some practical challenges, this powerful tool has been recommended as a complement to standard behavioral tests and to refine our management of pain, as well as to facilitate the implementation of humane endpoints [155,156]. Interestingly, the latter have been historically interpreted as a time point to perform euthanasia [157]. While it may be correct in many cases, recent research has focused on refining the concept itself by introducing non-euthanasia interventions when the endpoint is met [158]. An extended period with analgesia or facilitating access to food and water for mice after a painful and invalidating procedure would prevent further decline in their state, avoiding premature euthanasia and the use of additional animals [158]. If death is finally unavoidable, refined methods have also emerged as alternatives to sacrifice by carbon dioxide inhalation [159]. Despite being the most commonly used practice these days, doubts about its humanness still linger, making the options of anesthetic overdose or anesthetizing prior to cervical dislocation preferable in order to avoid additional distress in the animals’ final moments [157].

#### 4.2.2. Refinement in Liver Research

Once an animal is included in a specific protocol of liver research, there are still several aspects to improve from their experience welfare-wise. As mentioned, it is undeniable that any in-vivo experimentation on complex and severe diseases would prompt adverse effects on animals ranging from mild discomfort to long-lasting and intense pain, depending on the model. Surprisingly, in the case of mice and rats, this aspect is considered to be insufficiently explored and poorly tackled [160,161]. Several circumstances are thought to contribute to this phenomenon: Insufficient evidence-based information to guide effective treatments, concerns about unwanted interaction of analgesics in the experimental outcomes or lack of specifications, and underreporting analgesia in projects and published articles stand out among others [161]. Although it may not be realistic to expect that all pain is going to be successfully managed, simple strategies, such as anticipation of pain, multimodal analgesia, or a close evaluation of suffering and response to therapy post-intervention, as mentioned, should significantly foster welfare [160,162,163].

Surgery models: In the BDL model, the surgical model of chronic liver disease, it is needless to say how critical the role of the surgeon is in order to avoid unnecessary damage and the consequent pain during the procedure. Until the second half of the XX century, for instance, severe infections or bleeding were common in animals subjected to this model [164,165]. This was likely caused by the challenges intrinsic to rodent surgery. In contrast to bigger species where there is a dedicated surgical space and the surgeon is usually assisted, rodent surgery is routinely performed alone and outside a sterile environment. Thus, the responsible for performing the surgery must induce, maintain, and recover the animal from anesthesia while ensuring crucial aseptic conditions, especially when operating the abdominal cavity [166,167]. Unfortunately, this rather common setup is still far from the standards for bigger species, indicating an important room for the refinement of surgical procedures in rodents. Luckily, microsurgery has advanced at a rapid pace during the last two decades, optimizing the protocols for BDL in terms of a reduction of vascular structures and bile duct manipulation, improvements in aseptic techniques, and optimization of the postoperative mice, which resulted in an overall reduction of tissue damage [168]. Earlier studies also implemented the use of prophylactic antibiotics to reduce the risk of infection and early mortality [168]. Nevertheless, with the increasing importance of intestinal microbiota in the gut-liver axis [169] and the aforementioned spontaneous infections during cirrhosis progression [122], the application of antibiotics has been limited to highly specific circumstances. Remarkably, nowadays, BDL is considered as a low-complication, highly reproducible model for cholestatic disease in mice, albeit it is skill-intensive [167]. Additional refinements have been made in the model of partial BDL, in which only certain lobes are affected by the cholestasis, implying that the unaffected lobes can be used as internal controls while also reducing systemic distress [170]. Despite these innovations, pain is an integral part of the BDL model. Not only the laparotomy but especially the biliary pressure and the inflammation caused by the ligation of the duct, are a source of significant chronic pain and distress [171]. This pain is usually associated with notable levels of morbidity and mortality [30,172] and is considered traditionally neglected as well [173]. In that sense, local lidocaine-based anesthesia, either topically or injected in combination with bupivacaine, could provide multimodal analgesia at the incision site during laparotomy [174] when used along inhaled anesthetics such as isoflurane [160,172]. Alternatively, the ketamine-xylazine cocktail can be used as IP anesthetic, although tailoring the dose is necessary when applied to mice with chronic liver disease [160,175]. If this cocktail is used, preventive butorphanol or metamizole have been explored as a pain-anticipation strategy, showing positive results in the control with analgesia perioperatively [176]. In order to address further postoperative pain, buprenorphine has been the systemic analgesic of choice since the inception of the BDL model, albeit its insufficient use [173,177]. Interestingly, the multimodal combination of buprenorphine and carprofen outperformed buprenorphine alone controlling pain in postoperative mice [163], which could represent a refined analgesic protocol in BDL. Finally, when external pain mitigation is not enough to alleviate discomfort in this model, given its aggressiveness, early humane endpoints have been suggested to reduce excessive suffering in these mice [178].Substance Administration: Regarding the CCL_4_-induced chronic liver disease model, it is considered less severe than the BDL model [171] but not exempt from distress nor mortality [144,179]. Notably, depending on the route of administration of the toxin, the progression of liver damage can be more aggressive, leading to key differences between experimental designs, in turn, leaving room for optimization and refinement [141]. Intraperitoneal administration is the most common treatment route for CCL_4_ [180] (also for dimethylnitrosamine [13] or dioxin [14]), which, unlike thioacetamide [12], cannot be administered in drinking water due to their chemical and safety attributes [29]. This administration, spanning from 6 to 12 weeks depending on the protocol, renders a reproducible fibrosis that does not fully progress into cirrhosis in most cases [141]. Additionally, the process requires an experienced handler although the injection is considered technically simpler than other methods, in order to avoid organ puncture, bleeding, and infections [142]. Conversely, when advanced stages of the disease are needed, oral gavage CCL_4_ represents a viable alternative [181] despite being considered to induce greater rates of mortality [29].While earlier studies showed molecules, such as isothiocyanate or dihydrocollidine, to be safer when inducing cirrhosis orally [180], recent ones demonstrate that refined dosing protocols, in conjunction with careful handling during gavage, not only reduced mortality rates but also led to a full development of cirrhosis with portal hypertension which is difficult to replicate in rodents [182]. Remarkably, this latter study suggests that misplaced installation of CCL_4_ might be responsible for most of the deaths during the protocol, which, in turn, implies that further improvements during oral gavage would benefit the model [182]. Indeed, this route of administration is deemed technically challenging. Besides aspiration, gastrointestinal tract irritation, or even rupture in the worst scenarios, are some of the adverse effects associated with oral gavage [142]. In addition to learning a proper technique, utilizing softer intragastric probes would reduce gastrointestinal irritation [183]. When dealing with struggling individuals, habituation to the process might reduce resistance and stress. As a matter of fact, it has been shown that handling and restraint could be sufficient as an acclimatization protocol replacing the sham gavage treatment [184]. Further strategies to reduce discomfort during the intragastrical administration include the coating of the probes with sweet solutions when the model allows it [185] or, in extreme cases, the use of mild inhaled anesthesia for sensitive or aggressive animals [186].Imaging techniques: As mentioned, one of the outcomes of both, the BDL and CCl4 model, is the induction of a fibrotic process [3]. As such, the evaluation of fibrous deposition in the mouse liver plays a central role in these protocols. In order to assess the effect of genetic models or drugs on fibrogenesis, for instance, the liver is usually studied post-mortem with different molecular and histological techniques. Consequently, if the research is aimed at understanding a dynamic process, several time points along the protocol are usually scheduled with their corresponding higher number of mice enrolled [187]. Attempts to implement liver biopsy in mice, the gold standard in humans, have been made, striving for a reduction of the animals used in these cases [188]. Still, the concerns about its safety in rodents and its welfare implications make non-invasive approaches more suitable for following fibrosis progression [189]. Computed tomography [190] and magnetic resonance imaging [191] have been used to study liver processes, such as regeneration or metastasis growth. Nevertheless, the complex machinery required for these methodologies has prevented the spread of its use. Ultrasound-based approaches, on the other hand, are considered simpler, low-cost, and efficient, being widely used in human cirrhosis diagnosis [189]. In particular, transient elastography has been recognized as a potent tool to assess liver stiffness, which is proportional to the grade of fibrosis [192]. In fact, recent advances in the methodology have dealt with the technical difficulties related to the size of the mice, facilitating further longitudinal studies and promoting refinement and reduction in liver fibrosis research [193].Blood collection: Another excellent refined system to follow degenerative diseases of the liver is through the use of biomarkers present in blood and other biological fluids [194,195]. Traditional evaluation of liver function is based on the serum levels of hepatic enzymes, such as alanine aminotransferase (ALT), aspartate aminotransferase (AST), or alkaline phosphatase (ALP), as readout of hepatocyte damage. Nevertheless, additional proteins are gaining importance as prognostic markers of survival like Albumin or Liver-type fatty acid binding protein [196]. Moreover, MicroRNAs have been shown to be reliable markers of disease progression, especially miR-122, which not only is liver-specific but also strongly correlates with necroinflammatory activity in the liver and fibrosis progression [197]. Recent strategies based on omic technologies have also contributed to the identification of novel molecular markers fined-tuned to specific liver injury models [198,199], such as Keratin-18 or Glutamate dehydrogenase, ultimately increasing the set of tools available to follow disease progression. This increasing pool of markers is thought to have better sensitivity and specificity compared with the classical ALT and AST, giving rise to the use of combinatory panels of makers, which helps to accurately detect and predict risk during liver disease progression in a less invasive and humane manner [194,200].When it comes to collecting the blood from which most of liver-relevant biomarkers are assessed, there are also key differences to keep in mind between the existing approaches. In terms of animal welfare and experimental outcomes, not only the collection technique alters the values of crucial parameters, such as ALT and AST [201], but as they require more manipulation or are more invasive, they also induce greater levels of distress in the animals. Although intracardiac puncture or collection from the vena cava render higher volumes of blood, these are usually employed secondary to other interventions, such as organ harvesting, hence refinement is only applicable to suffering prevention during the procedure through a proper anesthesia and analgesia. On the contrary, with the advances in bioanalytical techniques, novel systems that require as little as 10–50 µL to function have emerged, promoting the development of microsampling techniques and favoring the implementation of longitudinal studies [202]. There is increasing evidence that microsampling techniques provide similar results to traditional approaches [203,204]. As such, they have been employed to perform several extractions from the same mouse, enabling the study of dynamic processes, like pharmacokinetics or toxicokinetics, from fewer mice, fostering reduction along refinement [204,205]. Regarding the welfare of this approach, it is true that taking several samples from one animal could be distressing, nevertheless, skilled handlers with the appropriate route of collection would maintain stress at a minimum [203,205]. For instance, facial vein puncture or sublingual puncture produce lower levels of stress when compared to other techniques like retrobulbar sinus puncture [206,207]. The latter is so aggressive that it requires the use anesthesia, its use is proposed as terminal, and even its use is discouraged in some countries [208]. Other routes that have been shown to induce acceptable levels of distress when applied to serial sampling are the lateral tail vein [205] and the saphenous vein [204]. Although there is still research to be done given the novelty of this field, the use of serial microsampling has a promising niche in the study of DILI [209].Genetic models: Finally, in the intersection between reduction and refinement, we can find the use of genetically modified animals as potential tools to refine liver studies ranging from drug metabolism to liver fibrosis, among other pathologies [18,210]. These models, most commonly generated in mice, allow researchers to mimic and study human liver diseases without the need for complex surgeries or special diets in many of the cases, thereby reducing manipulation and the consequent additional distress in the experimental animal. For example, in the BALB/c.*Mdr2*^−/−^ model, mice lack a canicular phospholipid transporter leading to an imbalance of phosphatidylcholine transport to the bile, which causes sclerosing cholangitis and liver injury. These animals develop a spontaneous progressive chronic biliary liver disease [187] that has been used to study biliary fibrosis progression without the need to generate the traditional BDL model through surgery [211]. Other interesting examples can be found for liver fibrosis or NAFLD elsewhere [212,213]. Even though these models have been proven useful as a result of similarities found between human and murine genetics and physiology, it is crucial to acknowledge that the remaining differences still constitute a burden in translational research (Figure 1). In particular, this aspect has affected drug development the most, having over 90% of the molecules that showed positive results after preclinical studies discarded in later stages of the clinical trials because of different levels of toxicity [214]. Remarkably, differences at the level of liver metabolism are at the core of the majority of these incompatibilities (Figure 1). About 50% of the molecules that caused DILI in humans showed no toxicity in the animal models [215]. To tackle this phenomenon, genetically humanized mice, which express human xenobiotic receptors, drug-metabolizing enzymes, and transporters, were first developed with positive albeit limited results [216]. Consequently, in order to mimic better the physiology and microenvironment of the human organ, mice with a chimeric liver, in which the hepatocytes have been replaced by human ones, were established to replace drug metabolisms accurately in vivo [214,216]. Interestingly, these humanized models have been successfully used for the evaluation of novel therapies, substituting more complex organisms like primates [217], or for the study of human liver viral infections [218]. Although the benefits of these models are unparalleled, there are several caveats still to address. The repopulation of the liver by the human hepatocytes is not perfect (>85%), which needs to be considered in the pharmacokinetic studies. Additionally, for the human hepatocytes to survive in the mice, different levels of immunodeficiency, depending on the specific model, must be generated to avoid rejection [18,214]. The implications of this lack of a complete immune response in humanized mice are also to be accounted for when modeling human liver pathologies since innate and adaptive immunity are involved in most of them [18]. Regardless of this pitfall, which could potentially be circumvented with the infusion of human immune cells as well [219], several attempts with promising results have been made to model chronic liver diseases, such as NAFLD [220] or liver fibrosis [221]. Whilst this is an incipient field, and the numerous variables and components of humanized mice make them extremely complex, they are a powerful tool for biomedical research, contributing to the refinement of the models by mimicking human disease better, while contributing to reduction by limiting the number of non-humanized mice used in preclinical studies.

### 4.3. Reduction

Since 3Rs principles were established in the fifties, the definition of reduction has been gradually settled due to its complexity [43]. Most definitions focus on reduction at the experimental or research project level, but certain reduction strategies may take an indirect approach by altering the research methodology or by modifying factors that are not directly associated with scientific procedures [222]. Currently, definitions from the scientific community globally refer to the reduction in the number of experimental units in the most efficient manner with the purpose of producing relevant and robust results to answer the scientific question, provided that all possible replacement strategies are considered [43,222,223]. However, as it is described in Russel and Burch’s book, the main aim of the 3Rs is to diminish or remove inhumanity in the treatment of laboratory animals [25], and therefore, some of the alternative strategies in animal testing cannot be defined separately as one unique principle [224]. In this regard, a balance between lowering the number of animals, acquiring solid evidence, and avoiding unnecessary injury to individuals is urgently needed [225]. Therefore, the avoidance of animal re-use is necessary when the cumulative impact of pain and suffering outweighs the utilization of new animals, based on the EU Directive 86/609/EEC [222].

The reduction principle can be subdivided into three approaches [222]: Intra-experimental Reduction, Supra-experimental Reduction, and Extra-experimental Reduction [25,222]. Intra-experimental reduction is the approach most frequently used, and it focuses on the strategies related with individual experiments in order to elaborate a proper experimental design [25,222]. By contrast, Supra-experimental approach is independent of the experimental research context, and it is related to conditions and settings in which animal experiments are performed and periodically overlap with refinement [222]. Last but not least, the Extra-experimental approach of reduction is not related to animal experiments and consists of studying the advancement of experimental methods on a global scal, achieved by implementing standardized guidelines for harmonization, as well as by introducing novel research and testing strategies [25,222], which frequently include not only replacement strategies but refinement as well. Taking all these approaches into account entails a better comprehension of reduction definition. The current review focuses on reduction strategies separately related to each approach.

The experimental design: These days, experts in animal research claim for the necessity of including a new 6Rs approach (including robustness, registration, and reporting) since the majority of scientists are mostly concerned merely with animal welfare when the 3Rs principles are applied [226]. In this sense, the UK National Centre for the Replacement, Refinement and Reduction of Animals in Research (NC3Rs) has recently created the new NC3Rs Experimental Design Assistant (EDA) that promotes the new 3Rs principles for scientific value, facilitating both the advancement of robust study protocols and enabling the inclusion of timestamps in the resultant protocols, gaining reproducibility and reducing variance [226]. Consequently, during the preliminary experimental phase, scientists should receive counsel from bio-statistical experts for the purpose of methodological scheduling the experimental design since it is of utmost importance and will define the research outcomes [25,222,224]. It is well-reported that during the nineties, over 60 percent of the published papers exhibited statistical errors and this fact led to an increased number of animals required for answering the scientific hypotheses [224]. Some of the solutions given by animal researchers were not only doing comprehensive research in the literature on the matter but also an improvement of scientists’ training regarding high technological statistics and avoiding duplicate tests sharing positive as negative results with the scientific community [222,224].As for the experimental design, the key elements to consider are: The hypothesis, objectives, types of variables (quantitative or qualitative), controls needed (positive or negative), the nature of the study (exploratory or confirmatory), the effect size of a given treatment, types of errors, the levels of bearable errors, and the statistical tests, among others [223]. Moreover, during experiment execution, blinding and randomization are two more crucial factors to bear in mind [223]. Regarding blinding, since in some instances it is inevitable, some measures, in particular using anonymous codes for identification and changing independent personnel for treatment administration, are suggested so as to reduce researchers’ prejudices about an experimental condition [223]. As for randomization, randomized block experimental designs have risen in importance as an alternative to completely randomized designs since they are considered more powerful and convenient and, in comparison, a reduced animal number is required. These designs are focused on reducing systematic errors by creating separated blocks composed of an experimental unit for a given treatment, in which an animal cage, not a single animal, is exposed to a drug combination, dosage, etc., in order to find a treatment effect [223,227]. This concept includes possible different variables between the EU that can affect the statistical outcomes because of possible independent-treatment effects from matched ages and weight, up to refinement conditions, such as the cage position on the rack and stress and treatment duration of each block [223,227].Taking all of these delimited variables into consideration, in addition, using a proper estimator of the sample size per group of individuals results in powerful statistical evidence with a sufficient number of animals [223]. In order to attain the estimated size of the number of animals, certain resources serve as supportive assistance, including the online assistance organization, the NC3Rs EDA’s website, and the widely utilized Gpower 3.1 software of Düsseldorf’s University.Data quality in the study design: Contemporary approaches to reduction focus on experimental design mostly for creating reproducible and quality data with fewer animals. Considerable apprehension has been raised in recent years regarding reproducibility, as evidenced by reviews, which was able to corroborate results in merely six out of fifty-three preclinical cancer trials [222,228] (Rt. Some of the reasons were included inappropriate cell lines and mouse models [222]. To a lesser extent, some experts have already considered that microbiota by itself can represent a source of variation in terms of immunological response in rodent models, and although, unfortunately, it is inevitable, it is suggested to be considered both in the first steps of the experimental design and analyzing the results in comparison with standardized studies in different laboratories on the same model [229,230].Another aspect to take into account in the reduction in animal experimentation is the sex as a biological variable (SABV). It is undeniable that for over a century, males have been preferred as experimental animals due to the possible effects of female hormones, and this fact could have potentially introduced bias and compromised the representativeness of the obtained results [231]. This approach can affect the sample size and the reduction principle in different aspects. Firstly, it has been discussed that when sex does not have a clear impact on the treatment, only a small number of additional animals are needed in order to have an adequate number of males and females per treatment. By contrast, the possible interactions of sex with treatment’s outcomes exponentially increase the required sample size [231]. In pilot and exploratory studies, the use of a single sex is generally accepted because of a reduction of the sample size since their main purpose is to discover potential effects and not the sex impact. However, large cohort studies could have resulted in a waste of animals and other resources if treatment effects are sex-dependent [231]. In this sense, according to the hypothesis and mouse model, governing authorities should enforce absolute transparency in relation to the inclusion and exclusion of animals, particularly concerning the selection of animal gender [231]. Secondly, the use of a single sex entails a misuse of female littermates in the breeding. In this sense, there exists an evident issue of notable concern in breeding facilities, which often exhibit animals in stock generating a surplus that surpasses ten percent of the overall bred population, and this can be addressed by facilitating information on that stock in repositories aiming to achieve a more optimal balance in the production and consumption of experimental animals [222].Further aspects to be mindful of are the training and educational level, which are mandatory in the vast majority of European countries when it comes to animal experimentation. Outcomes’ variability can be disrupted when accredited but non-experienced researchers perform experiments with animals, creating stress to animals regarding refinement conditions such as handling [222]. Furthermore, pharmaceutical companies incorporate the Good Laboratory Practice (GLP) and Good Manufacturing Practice (GMP), which both contribute to reduction by limiting the occurrence of dubious results and the necessity for re-testing, as high-quality and dependable data are employed, with clearly defined protocols in standard operating procedures (SOPs) [222].Use of up-to-date techniques: In contemporary times, novel technologies, as independent approaches within experimental design, have surfaced and gained recognition within the scientific community, aligning with the objective of implementing the reduction principle. During the past two decades, select fields of biomedical research, such as studies from pharmacological companies focusing on toxicity assessments, production, and vaccine testing, were developed in order to decrease the used animal number [222]. Notably, in the context of toxicity testing, the implementation of standardized protocols based on reversed toxicology has facilitated the direct evaluation of chemical exposure levels, deviating from the conventional four-phase approach. Consequently, this paradigm shift has led to a diminished reliance on animal models. Another notable example pertains to the meticulous control of vaccine batch production, wherein emphasis is placed on biochemical and physicochemical tests instead of evaluating the final product. Noteworthy historical instances include the replacement of a considerable amount of animals with a unique egg yolk for polyclonal antibody production and the utilization of recombinant DNA in hormone production, particularly during quality assessment procedures [222].Additionally, due to the high social pressure on replacement of experimental animal models, the cosmetic industry has largely replaced their usage for in vitro techniques in mutagenicity, phototoxicity, and skin corrosion [232]. In vitro techniques related with the replacement are specified in the replacement section. Some of the most remarkable novelty techniques related to reduction are the use of longitudinal imaging and multi-omics technology, which boost the information collected per animal [193,233,234]. Another innovative method related with genetic expression is the well-known CRISPR-Cas9 system, which although not yet standardized in practice, could prevent the creation of undesired gender from breeding in the near future [234].Reduction principle in liver research: First and foremost, precisely selecting a liver damage model according to the scientific question is imperative for researchers in order to fulfill the appropriate study design and reduce the number of animals with utmost effort. Additionally, this choice must be thoroughly chosen alongside a plenitude of factors, such as rodent strain and gender since the abundant diversity regarding clinical and histopathological features of the disease is already acknowledged [29].Concerning liver injury models in rodents, over the years, as it has been described, the scientific community has specifically tailored different models in accordance with the stages of hepatic damage [11]. Nevertheless, although they are handy for disease study, some of these models, such as ALD models, cannot reproduce the features of human disease. What is more, NAFLD/NASH models tend to recreate the disease, but they are not equally reproducible and count with a high disparity between them (methionine- and choline-deficient diet, high fat diet, Western diet, etc., because of the heterogenous diet composition, mice strains and gender are usually reported [11] (Figure 1).

As for gender’s part, it is universally accepted to use males in experimental NAFLD models since more severe liver histology changes have been appreciated in HFD and MCD diet in comparison with females [11,235]. Currently, the fact that female and male livers are metabolically different due to related genes with sex-specific effects on hepatic metabolism and because of the protective role of estrogen in liver damage has been addressed [235,236]. For instance, there is substantial evidence related to the innate immune system that demonstrates that innate immune cells (Kupffer cells and liver neutrophils) from males lead to the promotion of liver inflammation and fibrosis, meanwhile macrophages from females exhibit a tolerant profile by anti-fibrotic tendencies [235]. In fact, not only is it generally accepted that males are more susceptible to liver tumor development than females, but also a higher tendency of NAFLD-HCC in men can be noticed, whose effect can be justified by IL-6 production of Kupffer cells and the inhibitory role of estrogens [235]. Nonetheless, it has been reported that female C57BL/6 mice subjected to a high-fructose diet exhibited comparable liver steatosis to males despite experiencing greater hepatic inflammation, and what is more, they tend to lose this protective effect upon undergoing ovariectomy [236]. In this sense, it has also been reported that during a menopausal stage, both female animals and women present similar clinical outcomes in liver disease [236]. Taking all this into consideration, the majority of studies accept sex and age as independent variables. However, researchers opened a heated debate in this sense: Therapeutic targets and treatment responses should be evaluated upon hormone effects in preclinical to epidemiological studies and clinical trials in the female population to accomplish personalized medicine [236]. This debate has the potential to result in the inclusion of a larger number of animals initially but fewer in the long run, as the effectiveness of the drug should also be tested in female patients during clinical trials.

As for advanced chronic liver disease (CLD) models, it is undeniable that CCl4 has emerged as a widely utilized experimental model for liver fibrosis induction [11,141]. This model might be believed to be standardized, in spite of that, the reality is quite different since nearly all laboratories show protocols with variations in terms of length of treatment, doses, and administration methods [141]. This lack of standardization of SOPs affects the outcome of experimentation and increases the number of animals used.

Therefore, a correct choice of mouse strain to be used is important when it comes to reducing the number of animals used, considering the main objective of our study. In CCl_4_ models, BALB/c inbred mice are more susceptible to fibrosis development in comparison with C57BL/6 and FVB/N to an even lesser extent. Despite this fact, C57BL/6 are preferably used because of their availability in genetic modifications [141]. Moreover, fibrogenesis’ extent can be modified by the frequency and duration of the treatment. In this sense, a scenario like human stage 3 fibrosis can be reproduced in C57BL/6, with three times per week during four weeks or twice per week during six weeks of CCl_4_ IP [141]. However, this administration technique is generally argued because of the lack of fibrosis’ advanced stage obtained in C57BL/6 [11]. Additionally, it is highly crucial to establish the endpoint of the experiment after the CCl_4_ treatment since immediate proinflammatory states should be observed 24–48 h after the last dose, while for settled fibrosis and cirrhosis states, tissue harvest must be performed after 2–4 and 6–8 weeks, respectively [141]. Recently, some authors have reported a combined model of CCl4 and HFD feeding for the development not only of advanced stages of fibrosis but also of HCC, recreating human disease’s stages according to adapted treatment in weeks [11,131]. In this regard, a clear hypothesis upon a proper study design should be engaged with a specific animal model.

## 5. Conclusions

Experimental disease models are still necessary to be able to understand the pathophysiology of each of them and to be able to investigate the development of new therapeutic targets. In liver diseases, animal models are used to understand the molecular pathways involved and to develop possible treatments in preclinical studies. As has been described, there are numerous complementary and replacement methods to reduce animal testing. Unfortunately, currently, they are still not enough to recapitulate the complexity of the liver as an organ and its relationship with the rest of the organism through the circulatory system. In replacement models, in addition to these limitations, the need for special equipment and the enormous costs must also be considered. Furthermore, there is a wide variety of causes in the field of liver cirrhosis, with molecular pathways involved in its evolution still unknown. Therefore, animal models seem essential to study the clinical entity of cirrhosis until equivalent in vitro methods become available. Moreover, the cellular components used in most of these in vitro methods come from animals, so animal donors would be required in any case, although it is true that researchers would be applying the principle of reduction by reducing the number of animals used. Animal replacement in the field of liver diseases is only the beginning, with the concepts of reduction and refinement of techniques gaining special importance as long as the use of animals is necessary.

## Data Availability

Not applicable.

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
