# Peer review of "The 3Rs in Experimental Liver Disease"

_animals, 2023, doi:10.3390/ani13142357_

Round 1

Reviewer 1 Report

In this paper the authors present a review on the application of the 3Rs Principle to the study of liver diseases. Validity and limitations of each "R" is presented, with a particular focus on different experimental questions related to liver diseases.

I found the paper to be excellent. Without any prejudice, the authors discuss in a very convincing and informed way, the pros and cons of alternative methods, giving the correct interpretation of the term "Replacement". The other two "Rs" are also illustrated, in the framework of th topic of the paper.

This kind of approach needs to be circulated because, in my opinion, is  the most proper way to discuss alternative to and within animal experimentation: focused, particular but with an underlying scientific and ethical appropriate general attitude.

I would only suggest few clarifications to the text. 

in the Introduction, I would include a reference to th webpage "Alures" of the EU Commission, which provides the numbers of animals used in EU Member States (lines 54-63). As it stands now, the text is rather vague and too general. Please, do provide the actual numbers.

In the Introduction, I would delete the sentence from line 79 to 83. Its not very clear and rather confusing.

In Part 2 ("The 3Rs from and....view") on line 125, do the authors refere to Animal Welfare Bodies? And again on line 150?

After these minor clarifications, the paper surely is ready for publication. I applaude the efforts of the authors.

Author Response

Comments and Suggestions for Authors

  • In the Introduction, I would include a reference to the webpage "Alures" of the EU Commission, which provides the numbers of animals used in EU Member States (lines 54-63). As it stands now, the text is rather vague and too general. Please, do provide the actual numbers.

Following reviewer suggestion, the text of lines 54-63 has been rewritten, providing the actual numbers of the most used animal species in research, according to the last report of the EU Commission.

  • In the Introduction, I would delete the sentence from line 79 to 83. Its not very clear and rather confusing.

The authors agree with the reviewer that this sentence is confusing and has been removed from the text (line 79 to 83).

  • In Part 2 ("The 3Rs from and....view") on line 125, do the authors refer to Animal Welfare Bodies? And again on line 150?

According to the reviewer’s suggestion, a new text has been added (new line 175) to clarify this section.

Reviewer 2 Report

This is an important manuscript that highlights the critical importance of the 3Rs in experimental design and execution for research in experimental liver disease. The authors give a review of models used, with an emphasis on the value of the 3Rs as part of this process.  Given the emphasis on the 3Rs and their utility, if the authors could clarify a few points outlined herein, and provide important balance it would strengthen their conclusions.

Ln 15-17 –the paper weights the advantages and limitations of non-animal alternatives, as a result the authors conclude that the animal models are still required to address questions that face the field.  Consider revising the simple summary with the perspective of balance and the utility of the model to fit the scientific question of interest.

Ln 18-19 – “Experimentation in animal models is essential to understand the pathogenesis of human diseases and, considering the high prevalence of liver disease worldwide, to understand the pathophysiology of disease progression, and to design effective treatments.” Seems like this line is missing what is unique about animal models that enables this which would be easy to add to qualify this statement (e.g. integrated physiological systems).

Ln 22 – “However, today there is a growing awareness about the sensitivity and suffering of animals, causing opposition to animal research among society and many scientists.”  The authors should take care not to give the misperception that attention to animal welfare is a *growing* awareness as this has been built into regulations for decades in the majority of nations performing research using animals. The ability to replace animals or reduce there use has been more recently successful owing to new technology if anything.

Ln 35-78 – Excellent description by the authors of medical need.

Line 79 – “It has been necessary the arrival of a worldwide pandemic, to show the population  that they must become aware of the work of research.” This seems like it needs qualification or rephrasing.

Line 83 – “Already in 1959, Russell and Burch[25] developed what is today the basis of good research practice: The concept of the 3Rs (reduce, replace, and refine) to reduce the number of animals used, and to look for an alternative for total or if not partial replacement.” Good Research Practice (like GLP, GMP, or GCP) has a very specific meaning that relates to conduct and reporting of research not specific to animals, to which the 3Rs is not the basis, but rather contributes to ethical standards in animal research.

Figure 1 – very well done.

Ln 132-134 “European legislation” – Is the audience for this work only in the EU? Suggest that the majority of countries incorporate this legislation.  ‘Ethics committees’ can also be generalized for a global audience.  Considering the audience is beyond the EU should be throughout the manuscript.

Ln 141-142 “Current European regulations do not require a biostatistician to be part of an ethics committee, but the existence of this figure should be mandatory as it is key to correctly applying the r for reduction.”  This statement should be referenced.

Table 1 = excellent, considered including key associated references in the table if not visually detracting.

Thoughtful review of the various alternatives, just enough depth for the reader to grasp the message but not overdo.

Ln 602 – “loneliness-driven” – revisit this terminology. What the authors are describing is stress associated with singly housing a social species.

Ln 774 – “Blood extraction” – extraction is not typically used to describe simple blood ‘collection’ that is outlined in this section.

Ln 796 – “Although, intracardiac puncture or collection from the vena cava render higher volumes of blood, these are usually employed secondary to other interventions such as organ harvesting, hence there is scarce room for refinement.”  This line might give a reader who is not familiar with this technique the idea that this is done without proper analgesia which is a requirement for this type of sampling, such that pain or distress is fully managed.

In general, the authors have done a very comprehensive job distilling the 3Rs and work that has been done to expand these concepts more generally.This is a highly important topic that the authors have done an exceptional job in developing a comprehensive review of the application of the 3Rs in the field. With the minor considerations to even further clarify a few points, I congratulate the authors on this work.

Author Response

Comments and Suggestions for Authors

This is an important manuscript that highlights the critical importance of the 3Rs in experimental design and execution for research in experimental liver disease. The authors give a review of models used, with an emphasis on the value of the 3Rs as part of this process.  Given the emphasis on the 3Rs and their utility, if the authors could clarify a few points outlined herein, and provide important balance it would strengthen their conclusions.

  • Ln 15-17 –the paper weights the advantages and limitations of non-animal alternatives, as a result the authors conclude that the animal models are still required to address questions that face the field. Consider revising the simple summary with the perspective of balance and the utility of the model to fit the scientific question of interest.

According with the reviewer, the simple summary has been modified to show that the use of animal models, although necessary, can be reduced to answer the scientific question of the study.

  • Ln 18-19 – “Experimentation in animal models is essential to understand the pathogenesis of human diseases and, considering the high prevalence of liver disease worldwide, to understand the pathophysiology of disease progression, and to design effective treatments.” Seems like this line is missing what is unique about animal models that enables this which would be easy to add to qualify this statement (e.g. integrated physiological systems).

The authors agree with the reviewer, and a sentence has been added explaining why animal models are essentials to understand the pathogenesis of diseases.

  • Ln 22 – “However, today there is a growing awareness about the sensitivity and suffering of animals, causing opposition to animal research among society and many scientists.” The authors should take care not to give the misperception that attention to animal welfare is a *growing* awareness as this has been built into regulations for decades in the majority of nations performing research using animals. The ability to replace animals or reduce there use has been more recently successful owing to new technology if anything.

Authors agree with the reviewer that the sentence may cause controversy, and a statement has been added about the regulations in animal research.

  • Ln 35-78 – Excellent description by the authors of medical need.
  • Line 79 – “It has been necessary the arrival of a worldwide pandemic, to show the population that they must become aware of the work of research.” This seems like it needs qualification or rephrasing.

The authors agree with the reviewer that this sentence is confusing and has been removed from the text.

  • Line 83 – “Already in 1959, Russell and Burch[25] developed what is today the basis of good research practice: The concept of the 3Rs (reduce, replace, and refine) to reduce the number of animals used, and to look for an alternative for total or if not partial replacement.” Good Research Practice (like GLP, GMP, or GCP) has a very specific meaning that relates to conduct and reporting of research not specific to animals, to which the 3Rs is not the basis, but rather contributes to ethical standards in animal research.

Authors agree with the reviewer, and the sentence has been modified to clarify this fact.

  • Figure 1 – very well done.

  • Ln 132-134 “European legislation” – Is the audience for this work only in the EU? Suggest that the majority of countries incorporate this legislation. ‘Ethics committees’ can also be generalized for a global audience.  Considering the audience is beyond the EU should be throughout the manuscript.

As suggested, a new paragraph has been added to generalize the text for a global audience.

  • Ln 141-142 “Current European regulations do not require a biostatistician to be part of an ethics committee, but the existence of this figure should be mandatory as it is key to correctly applying the r for reduction.” This statement should be referenced.

According to the reviewer suggestion, he sentence has been modified

  • Table 1 = excellent, considered including key associated references in the table if not visually detracting.

As suggested, key references for each replacement strategy have been added as an additional column at Table 1

  • Thoughtful review of the various alternatives, just enough depth for the reader to grasp the message but not overdo.
  • Ln 602 – “loneliness-driven” – revisit this terminology. What the authors are describing is stress associated with singly housing a social species.

Following reviewer suggestion, the term has been changed for a more appropriate sentence.

  • Ln 774 – “Blood extraction” – extraction is not typically used to describe simple blood ‘collection’ that is outlined in this section.

Following reviewer suggestion, the term has been replaced.

  • Ln 796 – “Although, intracardiac puncture or collection from the vena cava render higher volumes of blood, these are usually employed secondary to other interventions such as organ harvesting, hence there is scarce room for refinement.” This line might give a reader who is not familiar with this technique the idea that this is done without proper analgesia which is a requirement for this type of sampling, such that pain or distress is fully managed.

The authors agree with the reviewer suggestion. A sentence has been added to clarify that analgesia and anesthesia is required during this procedure as well.

Reviewer 3 Report

The authors present a review on 3Rs approaches in the field of experimental liver disease, particularly focusing on cirrhosis and its pathological correlations. The review covers an overview of models and techniques where 3Rs have been progressively implemented, as well as models and techniques that offer the potential for further implementation in the future - in this case with particular regards to replacement, even partial, using in vitro models. The review provides good insights on the 3Rs approach for researchers involved in the specific fields, and authors should be commended for this effort. It would still anyhow benefit from several improvements under different perspectives, including animal welfare and animal model description, and English language and phrasing amendment through extensive and more accurate proofreading.

Comment 1:

Line 54 and 55: Provide bibliography.

At least for Europe, please consider https://circabc.europa.eu/ui/group/8ee3c69a-bccb-4f22-89ca-277e35de7c63/library/10ad28d6-e17e-4367-b459-20883402cfcc/details   

NHP, dogs and cats account just for the 0.2%. Please, consider rephrasing this sentence to highlight hierarchy of use.

Comment 2:

Lines 74 and 75. Jaundice on pigmented animals/strains is not always immediate to detect. It would be worth mentioning easiness (or the contrary) to recognise jaundice based on e.g. mouse strain background thus different colours. Jaundice is more easily visible in albino strains, less visible on black or agouti strains, especially at early stage. Also worth mentioning that the increased intensity in urine colour, might be easily detectable in metabolic cages, or using white bedding (e.g. paper bedding); very difficult to detect a change in urine colour using standard Aspen bedding. If the authors have specific examples and tricks describing how to easily detect such changes with standard bedding, they are strongly encouraged to describe them in the paper. It would be a valuable addition, to help implementing efficacy of daily checks thus refinement through identifying early endpoints, or specific time points to increase check frequency.

Comment 3: 

Lines 83 to 86: Already in 1959, Russell and Burch developed what is today the basis of good research practice: The concept of the 3Rs (reduce, replace, and refine) to reduce the number of animals used, and to look for an alternative for total or if not partial replacement. 

Include a brief mention of the purpose of Refinement - it is completely neglected from the sentence despite referring to 3Rs. Also, it is better to avoid capitalizing the letter (The) after the colon.

Comment 4:

Line 163: the authors state that due to mice small size, a high degree of technical expertise is needed as surgery is more complicated than in bigger models. Please note that, so phrased, this is a very slippery concept under veterinary good practices, welfare and science integrity principles too - suggesting that in other models a low degree of technical expertise is appropriate. High degrees of technical expertise are always needed and should be preliminary assessed and then required, when dealing with surgical procedures on any experimental model - independently from the species, the size of the animal and the complexity of the surgery. Please, rephrase the sentence accordingly.

Comment 5:

Lines 179 to 181: authors state that male animals are generally used due to a constant growth rate which enables a better health monitoring

Growth curves and rates of several rodent models are systematically updated and made available from Vendors' websites. Growth is constant in both sexes, but clearly on a different size range.

To justify the above sentence, please provide relevant bibliography in support.

It is also unclear the relation between growth rate and better health monitoring. Please clarify this further.

In the same lines, sex bias seems almost to be justified: male animals are generally used (...) to eliminate potential confounding factors such as the complex female hormonal status. 

Sex differences are clearly existing, nevertheless the research community as well as Funding agencies endorse that such differences should be overcome by including both sexes in the experimental design, and not at all selecting males only (or females only) just due to the reasons reported. Exceptions are strictly related to gender-specific illnesses.

I recommend to rephrase, clearly highlighting the intrinsic potential bias of such a choice, and then leave the sex bias discussion as presented in Lines 939-961.

Please refer to Plevkova, J., Brozmanova, M., Harsanyiova, J., Sterusky, M., Honetschlager, J., & Buday, T. (2020). Various aspects of sex and gender bias in biomedical research. Physiological research, 69(Suppl 3), S367–S378. https://doi.org/10.33549/physiolres.934593

Comment 6:

Lines 610 to 613: Authors state that tail tattooing is a refinement when compared to ear marking.

Despite the paper from Gaskill et al (2017), this topic is still very controversial, under the welfare perspective and in the light of refinement. Albeit advantages of tail tattoos in terms of reliability of identification, this practice is also aversive, causing significant tail inflammation and more agitation and anxiety that ear tagging.

I recommend to include this consideration in the sentence too.

Please refer to: Roughan, J. V., & Sevenoaks, T. (2019). Welfare and Scientific Considerations of Tattooing and Ear Tagging for Mouse Identification. Journal of the American Association for Laboratory Animal Science : JAALAS58(2), 142–153. https://doi.org/10.30802/AALAS-JAALAS-18-000057

Gaskill, B., Stottler, A., Garner, J. et al. The effect of early life experience, environment, and genetic factors on spontaneous home-cage aggression-related wounding in male C57BL/6 mice. Lab Anim 46, 176–184 (2017). https://doi.org/10.1038/laban.1225

Comment 7:

Line 682-690: Authors provide an unfortunately accurate description of routinary surgical approaches in rodents, outside a sterile environment, with a single operator struggling to maintain aseptic conditions when performing major surgeries, involving the opening of a body cavity. 

I strongly recommend to openly underline how the above described setting is a low-standard approach, certainly not enhancing animal welfare and Refinement, to bridge then with following sentence on Line 685 presenting the positive impact of microsurgery under this perspective.

Comment 8:

Lines 809 to 812: For instance, facial vein puncture produces lesser levels of stress when compared to other techniques like retrobulbar sinus puncture or sublingual puncture[201, 202]. The latter is so aggressive that requires the use of mild anaesthesia although its use is discouraged in some countries[203].

Actually, not only sublingual puncture does or may require anaesthesia. It is retroorbital puncture that is mostly recommended as terminal, non recovery procedure, always under anaesthesia (see NC3Rs reference below). Of note, bibliographic reference quoted by authors as number 203 (Whittaker et al, 2020) also refers to retroorbital samplings as a technique discouraged in some geographical areas (Australia). Please note that again Ref 203 is actually inconsistent with the first part of the sentence, as it clearly states that when compared together, sublingual sampling led to fewer traumatic lesions than facial. 

Please, rephrase the sentence correcting the sequence of techniques to have retroorbital sampling as the latter - to be consistent with bibliographic reference.

Please also see: Blood sampling: Mouse | NC3Rs

and Blood sampling: Rat | NC3Rs

Suggestion for improvement

Comment 9:

The review mentions humanized models, and the use of CrisprCas9 to potentially select for model gender as a tool for reduction, but lacks an accurate description of genetically modified animals used as models of liver disease, with a focus on Refinement specific approaches regarding GM models. If authors, based on their specific experience, would consider to implement a dedicated section/paragraph - including models description, accurate nomenclature of both GM strains and genes involved, refinement in breeding schemes and setting of strain specific humane endpoints - it would be a great addition to the manuscript.

Just as a matter of example:

Irie, J., Wu, Y., Wicker, L. S., Rainbow, D., Nalesnik, M. A., Hirsch, R., Peterson, L. B., Leung, P. S., Cheng, C., Mackay, I. R., Gershwin, M. E., & Ridgway, W. M. (2006). NOD.c3c4 congenic mice develop autoimmune biliary disease that serologically and pathogenetically models human primary biliary cirrhosis. The Journal of experimental medicine203(5), 1209–1219. https://doi.org/10.1084/jem.20051911 

010971 - NOD.c3c4, line 1112 Strain Details (jax.org)

Callegari, E., Domenicali, M., Shankaraiah, R. C., D'Abundo, L., Guerriero, P., Giannone, F., Baldassarre, M., Bassi, C., Elamin, B. K., Zagatti, B., Ferracin, M., Fornari, F., Altavilla, G., Blandamura, S., Silini, E. M., Gramantieri, L., Sabbioni, S., & Negrini, M. (2019). MicroRNA-Based Prophylaxis in a Mouse Model of Cirrhosis and Liver Cancer. Molecular therapy. Nucleic acids, 14, 239–250. https://doi.org/10.1016/j.omtn.2018.11.018

As mentioned in the previous section, extensive and very accurate proofreading is still required. The list below is indicative and non exhaustive.

Starting from the Title, and along the text (e.g. - but not limited to - lines 14, 63, 120, 122, 178...) the use of the saxon genitive is odd. Please correct 3R's into 3Rs along the manuscript, where needed. Note also that Line 174 reports only "R's", even without the 3.

Line 4: In the authors' names, remove "and" between surnames and name of the last author.

Line 49: his should be their as it refers to animal models.

Line 58: its small size, and also its low cost should be their, as the subject is still rodents from line 56. 

Lines 188 and 243: correct paragraph titles, probably automatically modified. From 4.1.1.2. D to 4.1.1. 2D in vitro liver models and the same for the 3D liver models.

Line 215: establishing, the e is missing.

Line 251: replace "is based in using" with "uses".

Line 562: the subject is plural (pathways plus knowledge), thus rephrase with "(...) keep computational models as a still-under-development technology. 

Line 690: correct "de use", "de risk" with correct article "the".

Line 711: correct "con" into "with" or "of" based on the desired sense of the sentence.

Line 807: and skilled handlers - correct either with "a skilled handler" or remove "and" to maintain the plural "skilled handlers". 

Lines 836 and 837: unparallel should be unparalleled; the are should be there are.

Line 842: to by accounted should be to be accounted for (...).

Sentence from line 847 to 851: numerous variables and component are plural, thus make and not makes. Please also replace the word regular referring to mice with a more informative adjective. 

Line 861: exhausted. I would recommend using a different, more informative word. It was not entirely clear to me without looking into bibliographic references if the sense here was "unavailable" or conversely "fully implemented".

Line 883: I assume "expertises" should be experts.

Line 915: EU refers to experimental unit? EU is also used for EU Directive; please consider extensive writing for experimental unit.

Line 1066: I assume "developed" might be "described", consistently with the sense of the sentence.

Lines 1069, 1070, 1073: "in addition" is repeated three times in very few lines. Please, consider rephrasing the entire paragraphs in a more organic way. Conclusion section would greatly benefit from a general rearrangement.

Line 1075: Since the article type is a review, I would suggest avoiding the use of "we" as strategies described derive from a global community effort and not from the Authors themselves. Please, consider using "researchers" or synonyms instead of "we". The same applies elsewhere (e.g. but not limited to Line 86) along the manuscript. Please, double check while proofreading the entire manuscript with accuracy.
